# Cadmium Exposure Is Associated with Behavioral Deficits and Neuroimmune Dysfunction in BTBR T^+^ Itpr3tf/J Mice

**DOI:** 10.3390/ijms24076575

**Published:** 2023-03-31

**Authors:** Mohammed M. Alanazi, Mushtaq A. Ansari, Ahmed Nadeem, Sabry M. Attia, Saleh A. Bakheet, Haneen A. Al-Mazroua, Abdullah A. Aldossari, Mohammed M. Almutairi, Thamer H. Albekairi, Marwa H. Hussein, Mohammed A. Al-Hamamah, Sheikh F. Ahmad

**Affiliations:** Department of Pharmacology and Toxicology, College of Pharmacy, King Saud University, Riyadh 11451, Saudi Arabia

**Keywords:** autism spectrum disorder, BTBR mice, cadmium, behavioral studies, Th17 cells

## Abstract

Autism spectrum disorders (ASD) are neurobehavioral disabilities characterized by impaired social interactions, poor communication skills, and restrictive/repetitive behaviors. Cadmium is a common heavy metal implicated in ASD. In this study, we investigated the effects of Cd exposure on BTBR T+ Itpr3tf/J (BTBR) mice, an ASD model. We looked for changes in repetitive behaviors and sociability through experiments. We also explored the molecular mechanisms underlying the effects of Cd exposure, focusing on proinflammatory cytokines and pathways. Flow cytometry measured IL-17A-, IL-17F-, IL-21-, TNF-α-, STAT3-, and RORγt-expressing CD4^+^ T cells from the spleens of experimental mice. We then used RT-PCR to analyze IL-17A, IL-17F, IL-21, TNF-α, STAT3, and RORγ mRNA expression in the brain. The results of behavioral experiments showed that Cd exposure significantly increased self-grooming and marble-burying in BTBR mice while decreasing social interactions. Cd exposure also significantly increased the number of CD4^+^IL-17A^+^, CD4^+^IL-17F^+^, CD4^+^IL-21^+^, CD4^+^TNF-α^+^, CD4^+^STAT3^+^, and CD4^+^RORγt^+^ cells, while upregulating the mRNA expression of the six molecules in the brain. Overall, our results suggest that oral exposure to Cd aggravates behavioral and immune abnormalities in an ASD animal model. These findings have important implications for ASD etiology and provide further evidence of heavy metals contributing to neurodevelopmental disorders through proinflammatory effects.

## 1. Introduction

Autism spectrum disorder (ASD) is a neurodevelopmental condition that affects behavior, learning, communication, and social interactions [1,2]. The World Health Organization estimates that 1 in 160 children worldwide have ASD, but its prevalence is difficult to determine in low- and middle-income countries [3]. Although ASD pathogenesis remains poorly understood, its causation appears to be multifactorial, involving genetic and other biological factors acting in concert with environmental influence within a specific developmental window [4,5]. A growing body of evidence indicates that immune dysfunction is associated with ASD pathogenesis and development [6,7]. Neuroinflammation and immune cell activation have been linked to ASD [8,9]. Moreover, the signal dysregulation of transcription factors related to Th1, Th2, Th17, and T-regulatory cells is implicated in ASD severity [10,11], while elevated chemokine and chemokine-receptor levels are involved in ASD behavioral impairments [8,12].

The excessive activation of Th17 cells can cause inflammation and induce autoimmune disorders [13]. Maternal Th17 cells and subsequent effector cytokine IL-17A signaling play an important role in ASD pathogenesis [14,15]. Furthermore, animal studies have identified a strong association of ASD-like phenotypes with Th17 cells; for example, the antibody blockade of IL-17A prevented abnormal behaviors in mouse offspring [16]. Additionally, STAT3 signaling is linked to Th17 cell differentiation. This transcription factor has been associated with neurodegenerative disorders [17] and may mediate the effects of maternal immune activation on depression-like behavior in mice [18]. STAT3 and RORγt signaling is involved in ASD-like symptoms associated with maternal immune activation [16,19].

In addition to IL-17A, other cytokines have been implicated in ASD. One is IL-21, expressed in brain lymphocytes and neurons under different neuroinflammatory conditions [20]. IL-21 significantly increases during mouse brain injury [21] and is upregulated in ASD children [22]. Similarly, tumor necrosis factor-α (TNF-α) is a proinflammatory cytokine predominantly expressed in CD4^+^ T cells. Its levels were significantly increased in the brains of subjects with ASD [23].

Exposure to heavy metals in early childhood adversely influences neurodevelopment [24,25]. The heavy metal cadmium (Cd) is frequently found in food, water, and air due to run-off from the nickel–cadmium battery industry and chemical fertilizers [26]. In humans, Cd aggregates in the stomach, liver, lungs, and brain [27,28]. The brain is most sensitive to Cd, with accumulation causing neurodegeneration and neuroinflammation [29,30] through the heavy metal’s potent proinflammatory effects [31]. Indeed, the association between Cd exposure, ASD, and other neurodevelopmental disorders has been well-established [32,33,34]. It is reported that Cd has undesirable effects on cells, including cell-cycle progression, differentiation, proliferation, DNA repair, replication, and apoptotic pathways [35,36,37]. Hair Cd levels decreased in children with autism or pervasive developmental disorders [38].

The BTBR T+ Itpr3tf/J (BTBR) mouse model is well-suited to ASD research [39,40]. BTBR mice exhibit highly replicable behavioral and immune abnormalities that have also been observed in children with ASD [41,42], including repetitive self-grooming and changes to ultrasonic vocalizations under social settings [43,44]. BTBR mice also have elevated levels of proinflammatory mediators [45], chemokines, chemokine receptors, and transcription-factor expression in Th1, Th2, Th9, Th17, Th22, and T regulatory cells [11,41,45,46,47]. A recent study also observed increased inflammation and oxidative stress associated with hepatic mitochondrial dysfunction, steatotic hepatocytes, and marked mitochondrial fission in BTBR mice [48]. Since Cd exposure leads to pro-inflammatory effects through upregulating multiple pathways, we hypothesized that Cd exposure would exacerbate behavioral deficits and neuroimmune dysfunction in BTBR mice.

## 2. Results

### 2.1. Cd Exposure Alters Repetitive and Social Behaviors

In BTBR mice, Cd exposure increased the average number of buried marbles from control (saline-treated) amounts. Strain (BTBR > C57: F (1,36) = 108.9, *p* = 0.0001), exposure (Cd > no Cd: F (1,36) = 23.71, *p* = 0.0001), and exposure × strain interaction (F (1,36) = 4.355, *p* = 0.0499) all had a significant effect (Figure 1A). The amount of time spent self-grooming also differed by strain (BTBR > C57: F (1,36) = 264.7, *p* = 0.0001), exposure (Cd > no Cd: F (1,36) = 47.34, *p* = 0.0001), and exposure × strain (F (1,36) = 18.58, *p* = 0.0003) (Figure 1B). These findings strongly indicate that Cd exposure negatively affects repetitive and social behaviors.

Our three-chamber social approach test results revealed that Cd-treated BTBR mice spent less time in the chamber with the stranger mouse than saline-treated BTBR mice (Figure 1C,D). Strain (BTBR < C57: F (1,36) = 119.0, *p* = 0.0001), exposure (Cd < no Cd: F (1,36) = 24.52, *p* = 0.0001), and exposure × strain (F (1,36) = 8.685, *p* = 0.0080) all exerted a significant effect on the social proximity index (Figure 1C). Likewise, both variables and their interaction strain (BTBR < C57: F (1,36) = 177.2, *p* = 0.0001), exposure (Cd < no Cd: F (1,36) = 23.71, *p* = 0.0001), and exposure × strain: F (1,36) = 7.986, *p* = 0.0104) significantly altered social domain exploration (Figure 1D). Our results confirmed that Cd exposure worsened the severity of social-behavior deficits in BTBR mice.

### 2.2. Cd Exposure Increases IL-17A/IL-17F Cytokine Expression

Flow cytometry revealed that Cd-treated BTBR mice had significantly more spleen CD4^+^IL-17A^+^ and CD4^+^IL-17F^+^ cells than saline-treated BTBR mice (Figure 2A,B). Strain (BTBR > C57: F (1,20) = 82.18, *p* = 0.0001), exposure (Cd > no Cd: F (1,20) = 17.94, *p* = 0.0004), and exposure × strain (F (1,20) = 6.670, *p* = 0.0178) all affected CD4^+^IL-17A^+^ cell count (Figure 2A). Likewise, the number of CD4^+^IL-17F^+^ cells was higher in BTBR mice than in C57 mice. Strain (BTBR > C57: (F (1,20) = 70.80, *p* = 0.0001) and was also higher in Cd-treated vs. saline-treated mice of both strains, exposure (Cd > no Cd: (F (1,20) = 16.57, *p* = 0.0006). The exposure × strain effect was significant (F (1,20) = 6.674, *p* = 0.0178) (Figure 2B).

Next, we assessed IL-17A and IL-17F mRNA levels in the brain. In BTBR mice, Cd exposure increased IL-17A and IL-17F mRNA expression more than saline alone. Strain (BTBR > C57: F (1,20) = 173.6, *p* = 0.0001), exposure (Cd > no Cd: F (1,20) = 21.58, *p* = 0.0002), and exposure × strain (F (1,20) = 12.83, *p* = 0.0019) significantly affected IL-17A mRNA levels (Figure 2C). Similarly, strain (BTBR > C57: F (1,20) = 172.3, *p* = 0.0001), exposure (Cd > no Cd: F (1,20) = 27.81, *p* = 0.0001), and exposure × strain (F (1,20) = 14.74, *p* = 0.0010) altered IL-17F mRNA (Figure 2D). Overall, Cd exposure influenced cytokine expression in Th17 cells of BTBR mice.

### 2.3. Cd Exposure Upregulates IL-21 Expression

We found that BTBR mice exposed to Cd had more spleen CD4^+^IL-21^+^ T cells than saline-treated BTBR mice. Strain (BTBR > C57: F (1,20) = 61.13, *p* = 0.0001), exposure (Cd > no Cd: F (1,20) = 24.57, *p* = 0.0001), and exposure × strain (F (1,20) = 9.116, *p* = 0.0068) all had significant effects (Figure 3A). In line with these data, IL-21 mRNA expression was upregulated in Cd-treated BTBR mice compared with saline-treated BTBR mice. The main effects of strain (BTBR > C57: F (1,20) = 105.7, *p* = 0.0001) and exposure (Cd > no Cd: F (1,20) = 19.73, *p* = 0.0003) were significant, as was their interaction (F (1,20) = 14.20, *p* = 0.0012) (Figure 3B). These results demonstrate that Cd exposure increases IL-21 production in BTBR mice.

### 2.4. Cd Exposure Increases TNF-α Expression

TNF-α-expressing CD4^+^ T cells were significantly higher in BTBR mice than in C57 mice. Strain (BTBR > C57: F (1,20) = 92.61, *p* = 0.0001) and Cd-treated BTBR mice than in saline-treated BTBR mice, exposure (Cd > no Cd: F (1,20) = 20.77, *p* = 0.0002); the exposure × strain interaction (F (1,20) = 4.724, *p* = 0.0419) was significant (Figure 4A). These findings were confirmed with RT-PCR demonstrating TNF-α mRNA upregulation in mouse brain (BTBR > C57: F (1,20) = 193.7, *p* = 0.0001; Cd > no Cd: F (1,20) = 8.360, *p* = 0.0090; exposure × strain: F (1,20) = 6.234, *p* = 0.0214; Figure 4B). Thus, Cd exposure may exacerbate immune abnormalities in BTBR mice partially via upregulating TNF-α expression.

### 2.5. Cd Exposure Increases STAT-3 Expression

Flow cytometry revealed a substantial increase in spleen STAT3-expressing CD4^+^ T cells of Cd-treated BTBR mice versus saline-treated BTBR mice. Both main effects strain (BTBR > C57: F (1,20) = 100.6, *p* = 0.0001; exposure (Cd > no Cd: F (1,20) = 23.48, *p* = 0.0001), and their interaction (exposure × strain: F (1,20) = 17.68, *p* = 0.0004), were significant (Figure 5A). Results from RT-PCR confirmed flow cytometry data, with strain (BTBR > C57: F (1,20) = 93.07, *p* = 0.0001), exposure (Cd > no Cd: F (1,20) = 10.18, *p* = 0.0046), and exposure × strain (F (1,20) = 4.724, *p* = 0.0419) all significantly influencing STAT3 mRNA expression in the brain (Figure 5C). These data suggest that Cd exposure contributes to the upregulation of STAT3 signaling molecules in BTBR mice.

### 2.6. Cd Exposure Upregulates RORγT Transcription-Factor Signaling

BTBR mice had significantly more RORγT-expressing CD4^+^ cells than C57 mice. Strain (BTBR > C57: (F (1,20) = 125.1, *p* = 0.0001), exposure (Cd > no Cd: (F (1,20) = 36.23, *p* = 0.0001). The exposure × strain interaction was significant (F (1,20) = 14.27, *p* = 0.0007) (Figure 6A). Consistently, RORγ mRNA expression increased in the brain of Cd-treated BTBR mice compared with saline-treated BTBR mice. Strain (BTBR > C57: (F (1,20) = 18.07, *p* = 0.0004), exposure Cd > no Cd: (F (1,20) = 108.7, *p* = 0.0001). The exposure × strain (F (1,20) = 11.89, *p* = 0.0025) also had significant effects (Figure 6B). These results suggest that Cd exposure upregulates Th17 transcription factor signaling in BTBR mice.

## 3. Discussion

Toxic exposure during critical developmental periods contributes to ASD etiology and exacerbates its symptoms [49]. At the same time, Cd is an environmental toxin and appears to modulate immune cell development [50,51], but its involvement in ASD development remains uncertain. Multiple characteristics suggest that Cd plays a role; however, it easily crosses the blood–brain barrier and accumulates in the brain, impairing synaptic activity, neurotransmission, and cognitive function [52,53]. Cd also influences cytokine production under various experimental conditions [54,55], and exposure activates several immune cells that produce proinflammatory cytokines and chemokines [56]. For example, Cd augments IL-6 and IL-8 production in astrocytes by activating NF-κB and MAPK pathways [57]. Cd also increases COX-2 and ICAM-1 expression in cerebrovascular endothelial cells [58]. In the mouse brain, Cd persistently activates neuronal apoptotic-related protein markers [59]. In this study, we observed that Cd treatment significantly altered ASD-like behaviors in BTBR mice, increasing stereotypic, repetitive, and anti-social behaviors. The effectiveness of Cd in deteriorating neurobehavioral development and increasing behavioral deficits in BTBR mice (e.g., increased marble burying and asocial preference in the three-chamber test) imply that exposure to the heavy metal could exacerbate ASD.

IL-17A is associated with behavioral impairments in ASD, suggesting that peripheral inflammation influences neuronal development [8,60]. IL-17A expression and serum concentration are elevated in ASD children [61,62], while maternal IL-17A levels affect ASD-like phenotypes in the offspring [16]. Our prior research has also demonstrated that IL-17A expression is upregulated in patients with ASD and the BTBR mouse model [10,11]. Importantly, IL-17A administration has been shown to activate microglia and cause a cascade of ASD-related brain pathologies [63]. Our experiments here supported previous reports. Cd exposure in BTBR mice significantly increased IL-17A/IL-17F expression in both the spleen and the brain. This enhancement of proinflammatory IL-17A/IL-17F expression by Th17 cells may be partially responsible for the deterioration of ASD-like symptoms in BTBR mice.

Cd treatment also increased IL-21 expression and elevated the number of CD4^+^IL-21^+^ T cells in BTBR mice. IL-21 is important to Th17 cell polarization [64] and has been detected in the brain during neuroinflammation [20] and injury [21]. IL-21 has been directly linked to ASD and upregulated in children with ASD [22]. Our findings in BTBR mice thus supported existing research indicating that elevated IL-21 expression is associated with greater severity of ASD symptoms. Furthermore, our data support the proposed mechanism: Cd exacerbates inflammation through upregulating proinflammatory signaling.

The inflammatory cytokine TNF-α is an essential modulator of neurogenesis and plays a critical role in ASD [65,66], which has been linked to neocortical neurogenesis [67,68]. Multiple studies have demonstrated that TNF-α is elevated in children with ASD [23,69]. Other research has similarly reported increased TNF-α production in adults with ASD [70], including in their peripheral blood mononuclear cells [11] and B cells [71]. In line with these previous findings, Cd-exposed BTBR mice exhibited a marked increase in spleen CD4^+^TNF-α^+^ T cells and upregulated TNF-α mRNA expression in the brain.

Inhibiting the neuroinflammatory STAT3 signaling pathway [72] decreases abnormal behavior [19]. Enhanced STAT3 expression in the hippocampus has been implicated in neuronal dysfunction [73]. Previously, we reported that STAT3 expression was increased in BTBR mice and children with ASD [11,22,74]. Here, we found that Cd exposure significantly upregulated STAT3 expression in the spleen and brain of BTBR mice, suggesting that the pro-inflammatory effect of Cd exposure is associated with high STAT3 expression, in addition to the involvement of interleukins.

RORγt is critical for mediating many autoimmune diseases [75], and pathogenic CD4^+^ T cells expressing RORγt contribute to neurobehavioral disorders by triggering brain inflammation [76]. Hence, its suppression has promise as an effective treatment for neuroinflammation [77]. RORγt expression is significantly upregulated in children with ASD and BTBR mice [10,78]. Corroborating these previous findings, we showed that Cd-treated BTBR mice exhibited a significant increase in CD4^+^RORγt^+^ T cells. Furthermore, Cd exposure in BTBR mice significantly increased RORγ mRNA levels. Thus, Cd exposure likely worsens neuroinflammation during ASD through the immune imbalance caused by elevated RORγt production. While the similarities between the BTBR mouse model and ASD are well known, it should be noted that there are always limitations in an animal model trying to mimic a human disorder. In ASD, subjects often have co-morbid disorders that are difficult to mimic in BTBR mice. These limitations should be considered while using BTBR mice as a model of autism-like behavior. Further investigations are also necessary to elucidate the role of Cd in modulating Th1/Th22 and T regulatory cells.

## 4. Materials and Methods

### 4.1. Chemicals and Antibodies

Cadmium, phorbol 12-myristate 13-acetate, ionomycin, 2-mercaptoethanol, and RPMI-1640 media were purchased from Sigma-Aldrich (St. Louis, MO, USA). Phycoerythrin; fluorescein isothiocyanate; allophycocyanin; PE/Dazzle; APC-Cy7-labeled CD4; IL-17A, IL-17F, IL-21, TNF-α, STAT3, and RORγt-labeled antibodies; FcR blocking reagent, RBC lysing solution, fixation buffer, and permeabilization buffer were purchased from BioLegend (San Diego, CA, USA). Golgi-Plugs were purchased from BD Biosciences (San Jose, CA, USA). TRIzol reagent was purchased from Life Technologies (Carlsbad, CA, USA). SYBR Green PCR Master Mix and High-Capacity cDNA Reverse Transcription Kit were purchased from Applied Biosystems (Foster City, CA, USA). Primers were purchased from GenScript (Piscataway, NJ, USA).

### 4.2. Animals and Cd Administration

Eight-week-old BTBR T+ Itpr3tf/J (BTBR) and C57BL/6 (C57) mice were purchased from Jackson Laboratory (Bar Harbor, ME, USA) and housed in a controlled, specific pathogen-free environment (12 h light/dark cycles at 25 °C). Animals also had access to food and water ad libitum from the animal center of the College of Pharmacy, King Saud University, Riyadh, Saudi Arabia. The Institutional Animal Care and Use Committee approved the study protocol. Cd was dissolved in normal saline to prepare appropriate concentrations for all experiments (Sigma, St. Louis, MO, USA). Control BTBR and C57 mice were treated with saline alone, while experimental mice were orally treated through oral gavage with 5 mg/kg/day Cd in normal saline daily for six weeks (*n* = 6–10/group). Ten BTBR and C57 mice per group were used for behavioral studies, and six BTBR and C57 mice per group were used for flow cytometry and gene expression analysis. This dosage was selected based on previous studies [44,79].

### 4.3. Self-Grooming

Mice were scored for self-grooming behavior, as reported previously [44]. After a 10 min habituation in the test cage, the total duration of self-grooming per mouse was recorded by a well-trained staff member blinded to experimental conditions. Each observation session lasted 10 min. As previously described, the observer sat approximately 2 m from the test cage [43,46].

### 4.4. Marble-Burying Test

Marble burying was measured in a standard mouse cage with 20 green glass marbles placed on clean bedding (5 cm deep). Marbles were arranged in a 4 × 5 grid following previous methods [41,80]. Each mouse was allowed 30 min to explore and bury marbles freely. As described earlier, a marble was considered buried if bedding covered at least 2/3 of it [46,81].

### 4.5. Three-Chamber Test

The protocol for the three-chamber paradigm followed published descriptions [7,43]. The test mouse was placed in a non-glare Perspex box (22 cm × 60 cm × 22 cm) for 10 min to habituate with both left and right retractable doors lifted. After placing the appropriate stimuli in the two side chambers, the test began with removing the left and right retractable doors simultaneously. In the following 10 min, the subject mouse was allowed to explore all three chambers. Two independent, blinded observers recorded social interactions, as previously described [7,43].

### 4.6. Preparation of Mouse Spleen Cells

Spleens from different groups were removed, and the single-cell suspension was performed as previously described [11]. Spleen cells were isolated by smashing the tissue with stainless steel mesh in RPMI-1640 medium containing 10% FBS, 50 μM 2-mercaptoethanol (Sigma-Aldrich, St. Louis, MO, USA), and 1% antibiotic antimycotic solution (Gibco-BRL). Cells were collected by centrifugation at 300× *g* for 10 min and then resuspended with 3 mL red blood cell lysis buffer (Biolegend, San Diego, CA, USA). After incubation for 10 min at room temperature, the cells were centrifuged at 300× *g* for 10 min and suspended in RPMI-1640 medium.

### 4.7. Flow Cytometry 

Flow cytometry was used to evaluate IL-17A-, IL-17F-, IL-21-, TNF-α-, STAT3-, and RORγt-expressing CD4^+^ T cells in spleens. Following the published protocol [10], splenocytes were incubated with PMA/ionomycin (Sigma-Aldrich, St. Louis, MO, USA), followed by the addition of Golgi-Plug (BD Biosciences, San Jose, CA, USA). Next, cells were rinsed with washing buffer/PBS, collected, and stained with fluorescein isothiocyanate (FITC)-anti-CD4, allophycocyanin (APC)-anti-CD4, PE/Dazzle-anti-CD4, PE/Dazzle-anti-IL-17A, PE-anti-IL-17F, PE-anti-IL-21, PE/Dazzle-anti-TNF-α, PE/Cyanine7-STAT3, and PE-anti-RORγt fluorescent antibodies (BioLegend, San Diego, CA, USA). Cells were analyzed using the FC 500 Flow Cytometer and counted in CXP (Beckman Coulter, Brea, CA, USA).

### 4.8. Gene Expression

RNA was extracted from brain tissue using TRIzol (Life Technologies, Carlsbad, CA, USA), quantified [11], and used for cDNA synthesis with a reverse transcription kit. Quantitative RT-PCR was then performed using SYBR Green Master Mix (Applied Biosystems, Foster City, CA, USA). The primers were as follows: IL-17A Forward, 5′-GGACTCTCCACCGCAATGAA-3′ and Reverse, 5′-GGGTTTCTTAGGGGTCAGCC-3′; IL-17F Forward, 5′-ACGTGAATTCCAGAACCGCT-3′ and Reverse, 5′-TGATGCAGCCTGAGTGTCTG-3′; IL-21 Forward, 5′-CTCCAGCCTCAGTCTCCTCT-3′ and Reverse, 5′-TCTGCTTCAGCTTTCGAGCA-3′; TNF-α Forward, 5′-GGACTAGCCAGGAGGGAGAA-3′ and Reverse, 5′-CGCGGATCATGCTTTCTGTG-3′; STAT3 Forward, 5′-CCCCAGGAATAGGGAGGACA-3′ and Reverse, 5′-TGGTATTGCTGCAGGTCGTT-3′; RORγ Forward, 5′-AGCTGTGGGGTAGATGGGAT-3′ and Reverse, 5′-ATCCGGTCCTCTGCTTCTCT-3′; and GAPDH Forward, 5′-GCATCTTCTTGTGCAGTGCC-3′ and GAPDH Reverse, 5′-TACGGCCAAATCCGTTCACA-3′. The 7500 Fast RT-PCR System (Applied Biosystems, Foster City, CA, USA) was used for amplification. Gene expression was normalized using GAPDH as an endogenous reference gene.

### 4.9. Statistical Analysis

Data are presented as means ± SD, *n* = 6–10/group (10 mice per group were used for behavioral studies, and 6 mice per group were used for flow cytometry and gene expression analysis). Differences were determined using two-way ANOVA and Tukey’s post hoc correction for multiple comparisons. All analyses were performed in GraphPad Prism 5 (GraphPad Software, San Diego, CA, USA). Significance was set at *p* < 0.05.

## 5. Conclusions

In conclusion, this study demonstrated that Cd exposure exacerbates repetitive and social behaviors in BTBR mice. The mechanism is likely related to upregulated Th17 signaling and, thus, the increase in proinflammatory mediators. These results suggest a crucial etiological role for Cd in ASD pathogenesis, specifically through causing or aggravating neurobehavioral and immune dysfunction. Our findings contribute important insight that can benefit the design of improved therapies for ASD and other neuroimmune disorders following Cd exposure.

## Figures and Tables

**Figure 1 ijms-24-06575-f001:**
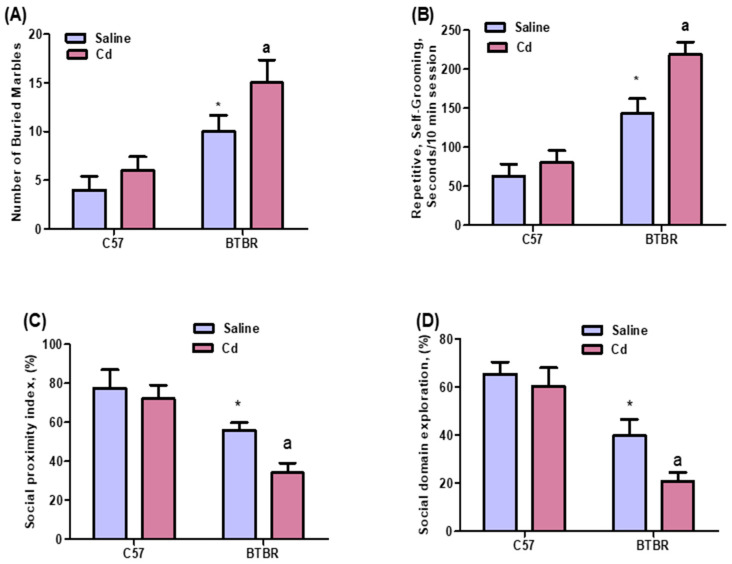
Effects of oral Cd exposure on social behavior in BTBR and C57 mice. Self-grooming (**A**). Marble burying (**B**). Social proximity index (**C**). Social domain (**D**). BTBR and C57 mice were exposed to Cd (5 mg/kg/day) daily for six weeks. Control C57 and BTBR mice received saline only. All data are shown as mean ± SD (*n* = 10/group). Different letter indicate significant difference. * *p* < 0.05 vs. saline-treated C57 mice, ^ap^ < 0.05 vs. saline-treated BTBR mice; two-way ANOVA with Tukey’s test.

**Figure 2 ijms-24-06575-f002:**
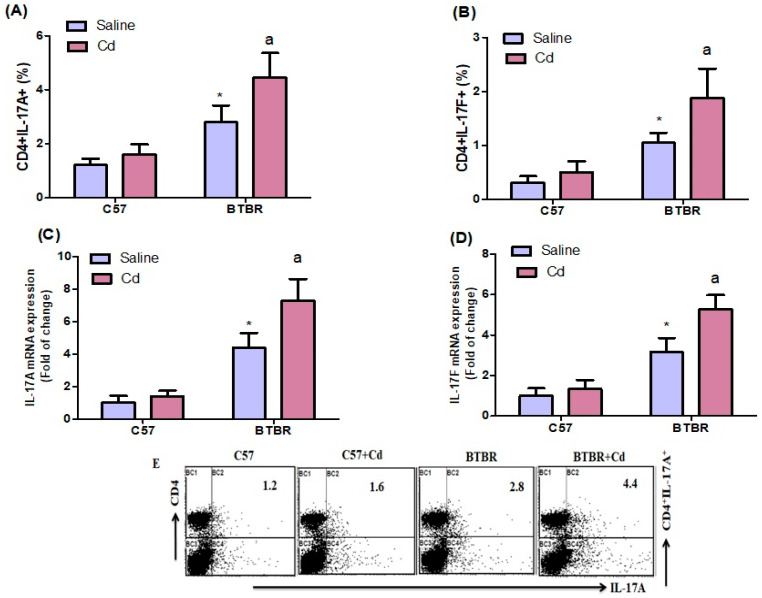
Effects of oral Cd exposure on IL-17A and IL-17F in BTBR and C57 mice. Spleen CD4^+^ T cells expressing (**A**) IL-17A and (**B**) IL-17F. RT-PCR analysis of IL-17A (**C**) and IL-17F (**D**) mRNA expression in brain tissue. (**E**) Representative FSC-SSC dot plots of CD4^+^IL-17A^+^ cells were obtained from each mouse spleen. BTBR and C57 mice were administered Cd (5 mg/kg/day) or saline only (control) daily for six weeks. All data are shown as mean ± SD (*n* = 6/group). Different letter indicate significant difference. * *p* < 0.05 vs. saline-treated C57 mice, ^ap^ < 0.05 vs. saline-treated BTBR mice; two-way ANOVA with Tukey’s test.

**Figure 3 ijms-24-06575-f003:**
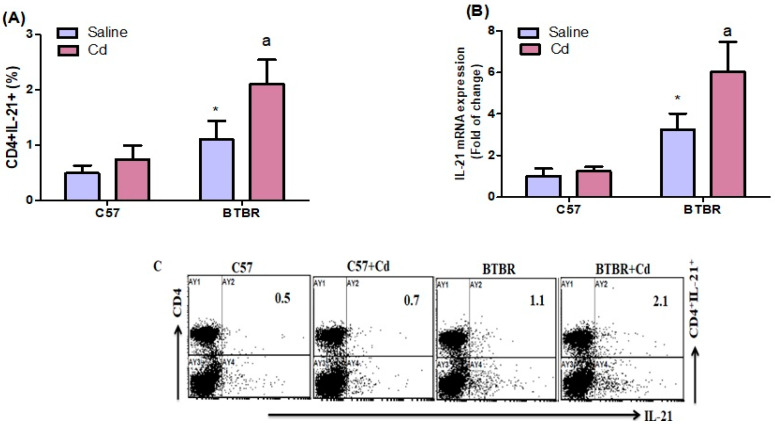
Effects of oral Cd exposure on IL-21. (**A**) IL-21-expressing CD4^+^ T cells from the spleen. (**B**) IL-21 mRNA expression in brain tissue. (**C**) Representative FSC-SSC dot plots of CD4^+^IL-21^+^ cells taken from each mouse spleen. BTBR and C57 mice were exposed to Cd (5 mg/kg/day) or saline only (control) daily for six weeks. All data are shown as mean ± SD (*n* = 6/group). Different letter indicate significant difference. * *p* < 0.05 vs. saline-treated C57 mice, ^ap^ < 0.05 vs. saline-treated BTBR mice; two-way ANOVA with Tukey’s test.

**Figure 4 ijms-24-06575-f004:**
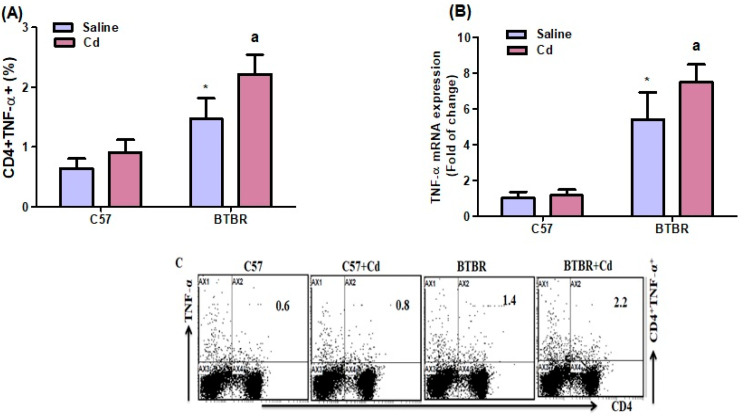
Effects of oral Cd exposure on TNF-α. (**A**) TNF-α-expressing CD4^+^ T cells from the spleen. (**B**) TNF-α mRNA expression in brain tissue. (**C**) Representative FSC-SSC dot plots of CD4^+^TNF-α^+^ cells taken from each mouse spleen. BTBR and C57 mice were orally administered Cd (5 mg/kg/day) or saline (control) daily for six weeks. All data are shown as mean ± SD (*n* = 6/group). Different letter indicate significant difference. * *p* < 0.05 vs. saline-treated C57 mice, ^ap^ < 0.05 vs. saline-treated BTBR mice; two-way ANOVA with Tukey’s test.

**Figure 5 ijms-24-06575-f005:**
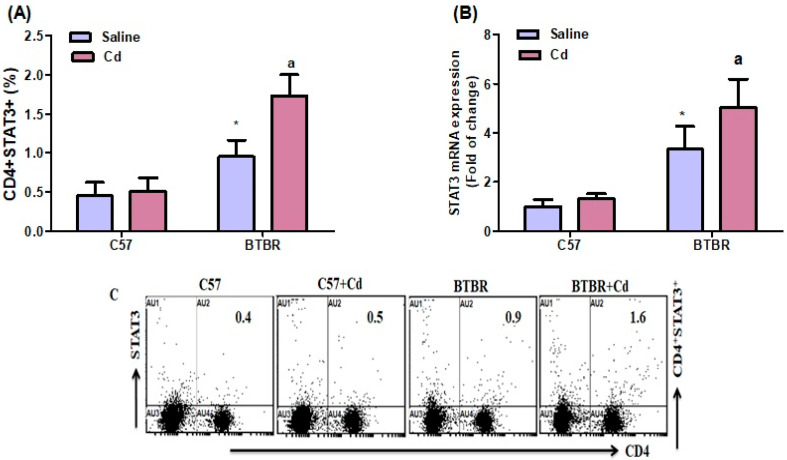
Effects of oral Cd exposure on STAT3. (**A**) STAT3-expressing CD4^+^ T cells from the spleen. (**B**) TNF-α mRNA expression in brain tissue. (**C**) Representative FSC-SSC dot plots of CD4^+^STAT3^+^ cells taken from each mouse spleen. BTBR and C57 mice were orally administered Cd (5 mg/kg/day) or saline (control) daily for six weeks. All data are shown as mean ± SD (*n* = 6/group). Different letter indicate significant difference. * *p* < 0.05 vs. saline-treated C57 mice, ^ap^ < 0.05 vs. saline-treated BTBR mice; two-way ANOVA with Tukey’s test.

**Figure 6 ijms-24-06575-f006:**
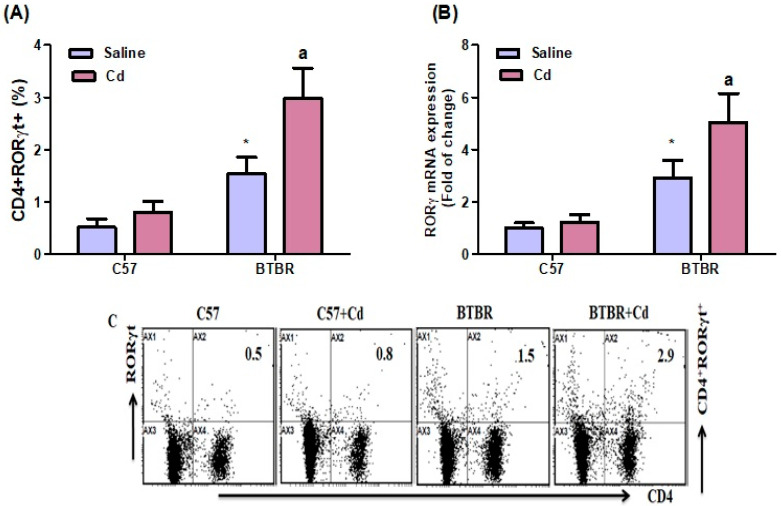
Effects of oral Cd exposure on RORγt. (**A**) RORγt-expressing CD4^+^ T cells from the spleen. (**B**) RORγ mRNA expression in brain tissue. (**C**) Representative FSC-SSC dot plots of CD4^+^RORγt^+^ cells taken from each mouse spleen. BTBR and C57 mice were orally administered Cd (5 mg/kg/day) or saline only (control) daily for six weeks. All data are shown as mean ± SD (*n* = 6/group). Different letter indicate significant difference. * *p* < 0.05 vs. saline-treated C57 mice, ^ap^ < 0.05 vs. saline-treated BTBR mice; two-way ANOVA with Tukey’s test.

## Data Availability

All data presented in this study are available on reasonable request from the corresponding author.

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
