# Peer review of "Cadmium Exposure Is Associated with Behavioral Deficits and Neuroimmune Dysfunction in BTBR T+ Itpr3tf/J Mice"

_ijms, 2023, doi:10.3390/ijms24076575_

Round 1
Reviewer 1 Report
The authors have done a good job in writing this manuscript, and the data obtained are really interesting. The introduction provides the reader with a valuable background on the issues discussed, although it could be improved. by implementing other aspects related to the correlation between high-fat diet consumption and cognitive disorders in both mother and offspring. It would be good to say a few words about the limitations of this study. The following are some comments and suggestions that I hope will improve the quality of the manuscript.
Line 64: Hepatic damage is also present in BTBR mice. The liver is a key organ in the body's detoxification processes, and therefore also for this reason, in ASD it might be more complicated to eliminate accumulated cadmium after exposure. This aspect should be added in the introduction, I recommend this paper to the authors: doi: 10.3390/antiox11101990.
Line 94: The dosage was selected from previous studies, what about the duration?
Line 117: Something is missing in this sentence.
Reviewer 2 Report
In the manuscript presented by Alanazi et al., the authors used BTBR T+ Itpr3tf/J (BTBR) mice as a ASD model and administered them with cadmium to assess its effect on behaviors and immune responses in BTBR mice. They found that cadmium exposure exacerbates repetitive and social behaviors in BTBR mice, which might be mediated by upregulating Th17 signaling and increasing proinflammatory mediators. The findings are interesting and of some significance. However, there are a significant number issues that need to be addressed.
1. The design of this study should be improved as no intervention was applied to inhibit the inflammatory or immune signaling pathway.
2. In the introduction section, the authors described ASD, the role of IL-17A and other cytokines in ASD, potential effects of STAT3 and RORγt signaling in ASD, and the association of cadmium exposure and ASD. However, some sentences need to be added between the paragraphs make them more clearly and logically organized. For example, “how cadmium affect ASD is still unknown and needs to be clarified” and be added in Line 63.
3. In Line 117, “…the single cell suspension was” is not a completed sentence.
4. Can the concentration of Cd in normal saline affect the result of this study? Are the concentrations the same between the groups? How was the saline administered to mice, through oral gavage or drinking freely?
5. In Line 163, “neuro behavior” is not a precise phrase. Marble burying test is used to assess anxiety-like behavior while self-grooming test represents a stereotyped repetitive behavior.
6. The total number of animals in each group should be mentioned in the Materials and Methods, Results, and Figure legends.
7. In Figure 6, A should be “RORγt-expressing CD4+ T cells from the spleen”, rather STAT3 mRNA expression.
8. The percentages of CD4+IL-17A+, CD4+IL-17F+, CD4+IL-21+, CD4+TNF-α+, CD4+STAT3+, and CD4+RORγt+ cells were pretty low in all groups although there were significant differences between groups. The low percentages of these cells may not have strong enough effects on the behaviors of BTBR mice.
9. In the Discussion section, the paragraphs were not logically organized and they were separated with each other.
Round 2
Reviewer 2 Report
Thank the authors for the revised manuscript, but there are still some issues that need to be clarified.
1. What does n = 6-10/group mean? The authors should write a precise number of animals used in each group.
2. A total number of BTBR mice and that of C57 mice should be mentioned in the Materials and Methods section.
3. The sentence "These findings strongly indicate that Cd exposure negatively affects behavior-related disorders" in Line 86 still need to be modified to clarify what kind of behaviors were affected by Cd exposure.
